# Application of Inositol Hexaphosphate and Inositol in Dental Medicine: An Overview

**DOI:** 10.3390/biom13060913

**Published:** 2023-05-31

**Authors:** Ana Druzijanic, Mare Kovic, Marija Roguljic, Livia Cigic, Martina Majstorovic, Ivana Vucenik

**Affiliations:** 1Department of Dental Medicine, University Hospital of Split, 21000 Split, Croatia; 2Department of Oral Medicine and Periodontology, School of Medicine, University of Split, 21000 Split, Croatia; 3Department of Orthodontics and Pediatric Dentistry, University of Maryland School of Dentistry, 650 W. Baltimore Street, Baltimore, MD 21201, USA; 4Department of Medical and Research Technology, University of Maryland School of Medicine, 100 Penn Street, Baltimore, MD 21201, USA

**Keywords:** phytic acid, molecular interactions, periodontitis, clinical observations, phytic acid, resolution of inflammation, tissue repair

## Abstract

Phosphorylated inositol hexaphosphate (IP6) is a naturally occurring carbohydrate, and its parent compound, myoinositol (Ins), is abundantly present in plants, particularly in certain high-fiber diets, but also in mammalian cells, where they regulate essential cellular functions. IP6 has profound modulation effects on macrophages, which warrants further research on the therapeutic benefits of IP6 for inflammatory diseases. Here, we review IP6 as a promising compound that has the potential to be used in various areas of dentistry, including endodontics, restorative dentistry, implantology, and oral hygiene products, due to its unique structure and characteristic properties. Available as a dietary supplement, IP6 + Ins has been shown to enhance the anti-inflammatory effect associated with preventing and suppressing the progression of chronic dental inflammatory diseases. IP6 in dentistry is now substantial, and this narrative review presents and discusses the different applications proposed in the literature and gives insights into future use of IP6 in the fields of orthodontics, periodontics, implants, and pediatric dentistry.

## 1. Introduction

Dentistry is a branch of medicine that mainly deals with the treatment of teeth disorders caused, among other things, by the action of bacteria.

Caries, for example, is the most common disease in the world, with an incidence of more than 50% in children and adults. If not corrected in a timely manner, deep caries leads to infection of the endodontic space and tooth loss, which ultimately affects chewing function and speech, compromises aesthetics, and leaves long-term psychosocial consequences that impact an individual’s quality of life [1].

Inositol hexaphosphate (IP6) is considered a “green”, natural molecule. It is the primary source of phosphate and inositol in edible plant seeds, legumes, and cereals, and the production of IP6 from rice grains is accessible and economically viable [2,3,4,5]. Except in plant seeds where it is found abundantly, IP6 is present in all mammalian cells, tissues, and bodily liquids, which makes it highly biocompatible [2,3,4,5,6,7,8] (Figure 1).

Although IP6 has been known since 1872, when it was discovered by Pepper, it has been studied in more detail only in recent years [5]. Its anti-inflammatory, antitumor, and antioxidant properties are based on a unique structure [2,6]. It contains a cyclohexane ring and six phosphate groups without direct phosphate–phosphate bonds, which is why it is highly negatively charged and has high affinity and chelating activity toward multivalent cations, such as calcium, magnesium, and iron [2,3,4,5,8,9,10,11,12]. The chelating effect of the multivalent cations generates hydroxyl radicals while preventing lipid oxidation and the formation of ROS—reactive oxygen species, molecules responsible for the degeneration of cellular functions and carcinogenesis. In addition, IP6 was also observed to have a significant effect on the chelation of oxygen-free radicals, preventing the exacerbation of chronic inflammatory diseases. At the molecular level, it has been shown to promote NK cell function and regulate the expression of the anti-inflammatory cytokines TNF-α and IL-β in neutrophils [2,3,4,10,12,13,14,15].

Because of its functions, IP6 is a potentially crucial dietary component, associated with preventing and suppressing the progression of chronic inflammatory diseases. In developed countries with a high prevalence of lifestyle-related diseases, the beneficial properties of phytic acids, such as biocompatibility, antitumor, antioxidant, and anti-calcification activity, are extremely important. Therefore, it is a promising compound that has the potential to be used in various fields of dentistry, including endodontics, restorative dentistry, implantology, and oral hygiene products, due to its unique structure and characteristic properties [5,10].

## 2. Beneficial Effects of IP6 for Human Health

IP6 is an essential molecule for various biological functions. The unique structure of six phosphate groups without direct bonds between phosphates gives it its characteristic effects [2,3]. The anti-inflammatory and antibacterial effects of IP6 have been demonstrated in numerous studies, and it has been shown to have a beneficial impact on suppressing chronic inflammatory diseases [3,7,15] (Figure 2).

Apart from the above effects on cations, IP6 also reacts with various enzymes, such as α-amylase, β-glucosidase, lipases, proteinases and other proinflammatory enzymes, starch, and many proteins [5]. IP6 has been shown to regulate the AMPK mediated energy expenditure pathway. AMP-activated protein kinase (AMPK) is a stress-activated protein kinase which is a major target in obesity and type 2 diabetes. In addition, AMPK stimulates glycolysis and coupled respiration-mediated ATP generation in metabolic tissues. The influence of IP6 on the AMPK pathway relies on inositol hexakisphosphate kinase (IP6K), an important mammalian enzyme involved in various biological processes such as insulin signaling and blood clotting. IP6Ks dephosphorylate IP6 to a distinct form of IP5, which modulates its targets by binding or by pyrophosphorylation, unlike IP6, which only regulates the cellular processes by binding to its protein targets. IP6Ks also regulate cellular functions via distinct mechanisms that do not require catalytic activity. For instance, IP6K1 interacts with glycogen synthase kinase and perilipin to modulate their cellular functions [16,17].

IP6 also contributes to storing minerals in the body, RNA transport, and DNA metabolism [13]. Omoruyi et al. have shown that the combination of IP6 and inositol as a food supplement may have a role in insulin secretion regulation, modulate serum leptin, and also decrease food intake, which leads to better control of food-associated weight gain, all of which can be contributory to better control of both prediabetic and diabetic states [18]. The antioxidant effect of IP6 is based on iron chelation, which catalyzes the formation of hydroxyl radicals while preventing the formation of molecules responsible for cell damage and carcinogenesis, such as oxygen free radicals [5,6,15].

IP6-generated inositol pyrophosphates increase glycolysis by boosting the mitochondrial functions, and consequently increase cellular energetics [15,19]. They also have a role in hemostasis; 5-IP7 is a key player in maintaining hemostasis. IPPs are major regulators of neutrophil function in infection and inflammation. Neutrophils have a major role in innate immunity and host defense, but the hyper-responsiveness and disproportionate accumulation can be harmful to the host, so neutrophils are under harsh regulation. IP6-kinase 1 regulates their function through 5-IP7-mediated spontaneous neutrophil death, which also helps in reducing the possibility of abnormal accumulation of neutrophils, and by doing so stops the generation of reactive oxygen species (ROS). IP6-generated IPPs also have a role in insulin secretion and sensitivity; IP6-kinase 1 has been shown to regulate insulin secretion and signaling, which can be targeted in type 2 diabetes therapy [20].

A number of studies suggest that IP6 has antioxidant, anti-inflammatory, and immune-enhancing capabilities. Moreover, Wee et al. have found a link between IP6 as a dietary component and the modulation of macrophage behavior through alteration of gene expression involved in pathways of pro- and anti-inflammatory responses, and resolution of inflammation pathways. According to their results, IP6 may represent a healthy diet to shape macrophage functions, which can lead to a beneficial impact on diverse diseases associated with uncontrolled inflammation [21].

Sclemmer et al. have found that IP6 also has an inhibiting effect on calcium salt formation, meaning it slows down formation of calcium oxalate kidney stones. They have shown a positive IP6 influence on glucose serum levels and cholesterol levels [3]. Presumed anticancer and therapeutic IP6 properties have been confirmed in other studies too. It affects several cancer-related critical molecular targets. Aside from enhancing immunity and antioxidant effects, which also contribute to tumor cell elimination, it too induces apoptosis and helps malignant cell differentiation return to normal phenotype, while reducing overall malignant cell proliferation [21].

Because of the above functions, IP6 is a potentially important dietary component associated with the prevention of chronic inflammatory diseases. Wee et al. also conclude that IP6 can be used as a dietary component to regulate acute inflammatory responses by acting on the expression of genes related to anti-inflammatory responses and by interrupting inflammatory signaling pathways in macrophages. Such an effect may have a positive impact on various diseases associated with an uncontrolled inflammatory response [9].

In addition to the described effects, Nassar et al. showed that IP6 has significant antimicrobial properties against biofilm cultures and exhibits a rapid bactericidal effect against *E. faecalis*. This study was the first to demonstrate the antimicrobial properties of bacterial biofilms, which can be investigated and exploited not only through applications in dentistry but also more generally [10].

As IP6 shares some properties with inorganic linear condensed polyphosphates typically used in oral hygiene products [11], it is suggested that it may affect the metabolism of *S. mutans* by modifying glycolysis enzymes and chelating essential metabolites. There is also evidence of sensitivity of some strains of *Lactobacillus* sp. and *Streptococcus* sp. to IP6 ingested in food [13].

The broad spectrum of antibacterial activity of IP6 and its rapid action suggest a bactericidal effect by destroying the bacterial membrane and a possible link with chelating activity towards iron ions. IP6 has biostatic and biocidal effects against various microorganisms, such as Gram-negative and Gram-positive bacteria, drug-resistant strains, and *C. albicans* [14]. Furthermore, Gan et al. showed how the surface IP6–metal complex shows a significant reduction in implant surface colonization by the Porphyromonas gingivalis [22].

These findings are of extraordinary importance for the prevention and treatment of various pathological conditions that lead to compromised oral health, such as caries, but also for the long-term durability of resin-based fillings, showing great potential and a possible broad usage of IP6 in different branches of dental medicine.

## 3. IP6 Anti-Nutrient Effects and Their Potential Impact on Dental Health

IP6 is considered an anti-nutrient due to its strong binding affinity with several mineral micronutrients, such as Fe, Zn, Mg, Ca, Mn, and Cd, which then affects their physiological metabolism and bioavailability [23]. Phytases enzymatically hydrolyze IP6 into different forms, out of which only the original form, IP6, and IP5 have the aforementioned inhibitory effect on mineral bioavailability. Out of the minerals mentioned above, only iron deficiency has an effect on oral structures. According to Rahman et al., iron deficiency leads to the development of atrophic glossitis, angular cheilitis, burning mouth syndrome, and oral ulcerations. Such deficiencies can be found in a pure vegetarian diet too, and using appropriate iron supplements can adequately increase the absorption of iron and improve the symptoms [24,25].

## 4. IP6 in Restorative Dentistry and Endodontics

The endodontic treatment procedure includes removing necrotic, infected pulp tissue from the endodontic canal and its preparation for filling with a biocompatible filling material with complete closure of the endodontic space. Canal preparation includes procedures for cleaning and widening the canal with irrigation using an agent that has multiple effects. For example, the ideal irrigant should destroy microorganisms, dissolve necrotic tissue, remove the residual layer, and penetrate inaccessible areas with low surface tension without affecting healthy tissue. Today, sodium hypochlorite (NaOCl) and ethylene diphosphate tetraacetic acid (EDTA) are most commonly used in modern endodontics as agents with the above properties.

IP6 is one of the most researched compounds with the potential to become a replacement for EDTA [15,26].

It is a natural, negatively charged molecule with the ability to remove the residual smear layer [8,23,27]. Because of its high negative charge, IP6 is a potent chelating agent, even towards multivalent cations, and it is precisely because of this property that it is being investigated as a biocompatible alternative in root canal irrigation [10,18,26].

It has been shown that the use of IP6 for final root canal irrigation during biomechanical treatment shows promising results very similar to those obtained with 17% EDTA, which is currently the most commonly used solution for removing the residual layer in the root canal [15]. IP6 successfully removes the residual layer with a mild etching effect that allows exceptional bond quality due to better infiltration of resin monomers while the depth of demineralization is low, which means minimal damage to healthy tooth tissue. In addition, IP6 appears to naturally cross-link collagen fibrils, making them more stable and less prone to collapse, resulting in preserved interfibrillar spaces that then allow for high-quality monomer impregnation [26,27,28]. Nassar et al. found that IP6 root canal rinsing reversed the harmful effect of NaOCl on dentin–resin adhesion without the side effects of using EDTA [29].

Ideally, the collagen matrix from the demineralized dentin should be fully embedded in the resin and also fully polymerized, creating a continuous network of collagen–resin bonds that ensures successful adhesion of the dental filling to the dentin [28]. The exact mechanism by which phytic acid increases the strength of the bond between fillings and dentin has not been fully elucidated, but it is thought to cross-link the network of exposed collagen fibers. Forgione et al. hypothesize that IP6 has the ability to form cross-links in exposed dentin collagen, as has already been shown in the case of protein cross-links, with seven times less hard tissue lost than when treated with phosphoric acid. This putative ability could positively influence the mechanical strength of the collagen [30,31].

Kong et al. concluded that during etching with IP6, there might be an interaction between IP6, collagen, and calcium ions in hydroxyapatite, forming a complex ternary IP6-collagen-Ca, which could strengthen the bond between dentin and resin and protect dentin collagen from degradation. Compared with EDTA, IP6 showed a better result in preventing the degradation of collagen by bacterial collagenases. The remaining smear layer was successfully removed, the dentin was etched, and the bond was strong and satisfactory, with minimal collagen degradation and nano-leakage [28]. In the study by Puvvada et al., it was shown that IP6, in combination with sodium hypochlorite, showed the best results in root canal irrigation [6].

As mentioned above, IP6 has a proven biostatic and biocidal effect on bacteria and bacterial biofilms, which is of great importance in endodontics and an essential factor in the success of root canal treatment since its antibacterial effect on the destruction of the bacterial membrane is immediate and, at the same time, is completely biocompatible. It is bactericidal against a wide range of Gram-positive and Gram-negative bacteria, including *E. faecalis*, a bacterium responsible for most periapical inflammation. It also acts on drug-resistant strains as well as on *Candida albicans* sp. [10,32].

Considering that it is a biocompatible and natural molecule, IP6 seems to be an ideal molecule for use in final root canal irrigation. Further studies are needed to shed light on the other properties of IP6, especially in clinical practice [15,27].

## 5. IP6 in Implantology

Dental implants have become one of the most commonly used materials to replace one or more missing teeth [33]. Modern implants are made of titanium alloy with different additives, depending on the implant manufacturer. The implantation process is standardized by a protocol.

Immediately after implant placement, it is necessary to promote osseointegration while providing adequate antibacterial protection. Recently, the use of bactericidal metal ions has made it possible to eliminate the need for antibiotics as additives on the implant surface, thus reducing the risk of antibiotic resistance in bacteria [23].

The modified surface should have the ability to inhibit bacterial adhesion while promoting the adhesion of somatic cells. Phosphate can be covalently bound to metal oxides on the surface. Therefore, IP6 has been used for years to treat magnesium and its alloys to improve corrosion resistance. Therefore, the use of phytic acid to modify the surface of dental implants has also been explored [7,13].

Recently, IP6 has been used as an additive in various substances to modify the implant surface. It is a fully biocompatible molecule with an antitumor effect and a high affinity for hydroxyapatite adhesion. Due to its strong binding affinity to multivalent metal ions, various types of phytic acid compounds have been developed with metals as modifiers for implant surfaces to improve their anticorrosive properties [7,22]. Due to its structure with a cyclohexane ring and six negatively charged phosphate groups, without direct phosphate–phosphate groups, it reacts directly with titanium to form titanium oxide without the need for a binder. In addition to titanium, its chelating capabilities provide direct contact with calcium and iron cations in the implant alloy. The result is a bioactive, functionalized surface with improved osseointegration and reduced resorption of surrounding bone, with reduced adhesion of bacterial biofilm [5,13,32].

Lun et al. have shown that the use of phytic acid in surface modification with calcium ions improves osseointegration and shows potential for use in a variety of applications, and outside of dental implants [13].

There is increasing evidence that the antioxidant, anti-inflammatory, and immunomodulatory properties of IP6 may increase bone density and reduce bone loss [6]. Gan et al. showed that the IP6 metal surface complex exhibited excellent antibacterial properties against Porphyromonas gingivalis and significantly reduced colonization of the implant surface. The abundance of phosphate groups significantly increases the hydrophilicity of the surface, promotes the early adhesion of proteins, and stimulates the proliferation of bone marrow mesenchymal cells. In addition, phosphate groups promote hydroxyapatite deposition by providing sites for chelating reactions with calcium ions [22].

These conclusions should prompt further research into the potential of IP6 to produce implants with higher quality and better long-term clinical outcomes [11].

## 6. IP6 in Oral Hygiene Products

Toothpaste and toothbrush are the most important means of maintaining oral hygiene. They contribute to the mechanical removal of plaque, are multifunctional, and help control diseases in the oral cavity (caries, gingivitis, dentin hypersensitivity, etc.) while providing cosmetic benefits and preventing bad breath [14].

An important function of toothpaste composition is to control the formation of stains caused by food, tobacco, or drugs that bind to the protein components of enamel pellicles. Currently, abrasive particles are used in almost all kinds of toothpaste to control staining. They use their physical properties (friction) to remove pigments during tooth brushing to counteract the formation of discoloration. It is known that electrostatic interactions associated with the adhesion of proteins to enamel and hydroxyapatite are essential in the formation of discoloration. For this reason, soluble acids are regularly added to toothpaste to disrupt the binding of proteins to hard tooth surfaces to provide additional stain control. Furthermore, adding chemical agents to conventional oral hygiene products attempts to improve their functions without increasing the abrasiveness of the paste, making it a cheap solution and an attractive additive [14,26].

Interest in the potential protective effects of IP6 for human health is not new. IP6 shares some similarities with inorganic linear condensed polyphosphates commonly used in oral hygiene products and is considered a possible substitute for them. IP6 is thought to affect the metabolism of *S. mutans* by modifying the glycosylation of enzymes. Evidence shows that ingested *Lactobacillus* sp. and *Streptococcus* sp. strains are also sensitive to IP6 [3,5]. There is evidence that the chemical action of sodium phytate causes discoloration to disappear from surfaces that are generally not well brushed during tooth brushing (lingual surfaces) and from hard-to-reach areas (interproximal surfaces). These hard-to-reach areas show the greatest reduction in extrinsic tooth discoloration [14,26].

Grases et al. have shown a statistically significant reduction in total calcium, magnesium, and phosphorus deposits during the use of an IP6 rinse, demonstrating the effect of such an oral hygiene protocol on reducing the amount of hard plaque. Phytate inhibits the crystallization of hydroxyapatite and brushite and can potentially prevent the formation of hard plaque. The presence of other metal ions such as zinc enhances the inhibitory effect of IP6 on the crystallization of calcium oxalate. They concluded that IP6 effectively reduces the amount of hard plaque and that such a mouth rinse could be a valuable treatment to prevent plaque formation [8].

The study by Creeth et al. showed that toothpaste containing cyclic phytate ions with 1150 ppm NaF has positive effects on remineralization and demineralization processes on the surfaces of early erosive enamel lesions [11]. IP6 is a potential replacement for the conventional linear condensed polyphosphates commonly used in modern toothpaste. It possesses most of the conventional polyphosphate properties, but unlike them, it does not interfere with fluoride bioactivity via the inhibiting mineral exchange on the enamel surface, as they do. In in vitro studies, IP6 modified ion transport from the enamel and dentin surfaces, potentially slowing the progression of caries. In addition, Parkinson et al. showed that neither IP6 nor zinc addition affected the ability of fluoride to promote enamel demineralization and remineralization processes [11,14]. When IP6 is added to toothpaste, it can successfully contribute to the removal of intrinsic stains and also prevent the formation of new stains without increasing the abrasiveness of the toothpaste, which is of great importance in patients with dentin sensitivity problems. The mechanism behind these effects is the ability of IP6 to bind to the enamel surface by chelating the calcium in hydroxyapatite and preventing the adhesion of proteins and bacterial pellicles to the tooth surface [5].

The cariostatic mechanism of IP6 has not been fully elucidated, but there are several possible mechanisms. Its local effect is considered more likely than a systemic effect. IP6 binds extremely rapidly to hydroxyapatite and forms a monomolecular layer on the surface of the crystal, resulting in increased resistance to the dissolution of the crystal by acid due to the ion diffusion barrier. Another suspected mechanism is the formation and precipitation of the calcium–IP6 complex on the crystal surface. Washing with water and mild acid does not decrease the amount of these complexes on hydroxyapatite, indicating a strong bond between IP6 and hydroxyapatite. Moreover, the adsorption of IP6 on hydroxyapatite causes a change in the surface charge of the crystal, preventing plaque accumulation by causing an adverse change in the affinity of bacterial proteins for tooth surfaces [5,8].

These findings make IP6 a very interesting addition to conventional oral hygiene products and should be thoroughly explored, especially considering the fact that IP6 is abundant in plant seeds and the economically viable and straightforward extraction procedure.

## 7. Conclusions and Future Directions

IP6 is a widely used natural phosphate and energy storage molecule. It is biocompatible and has various beneficial effects as a dietary supplement. Its occurrence and a feasible and straightforward extraction procedure, together with its properties, make it a fascinating and promising compound to be used in various fields of dentistry, including endodontics, restorative dentistry, implantology and oral hygiene products. No studies have shown any potential concerns with using IP6 in dentistry.

Further studies should be conducted to uncover the mechanisms behind all the beneficial effects of this exciting molecule and discover the possibility of using it in even more products and procedures in dentistry.

## Figures and Tables

**Figure 1 biomolecules-13-00913-f001:**
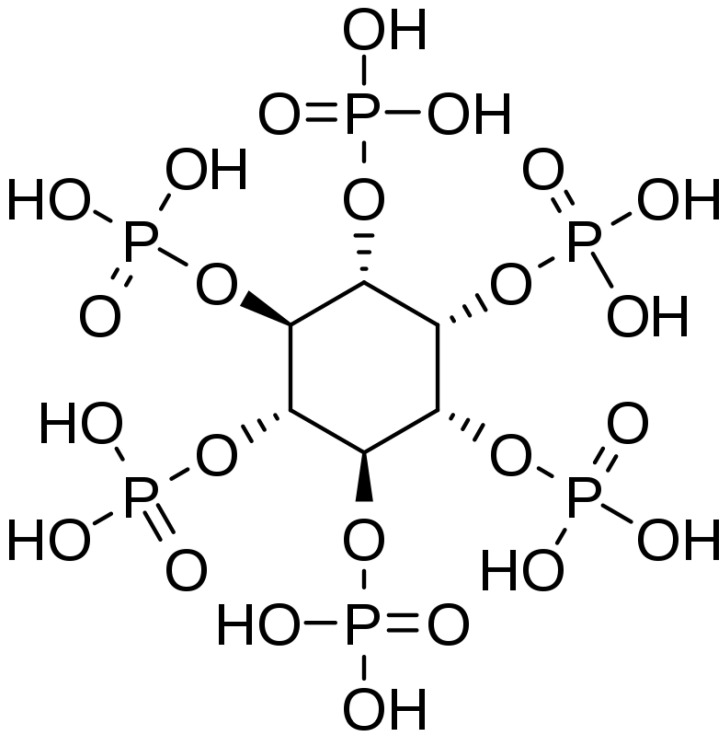
IP6 chemical structure.

**Figure 2 biomolecules-13-00913-f002:**
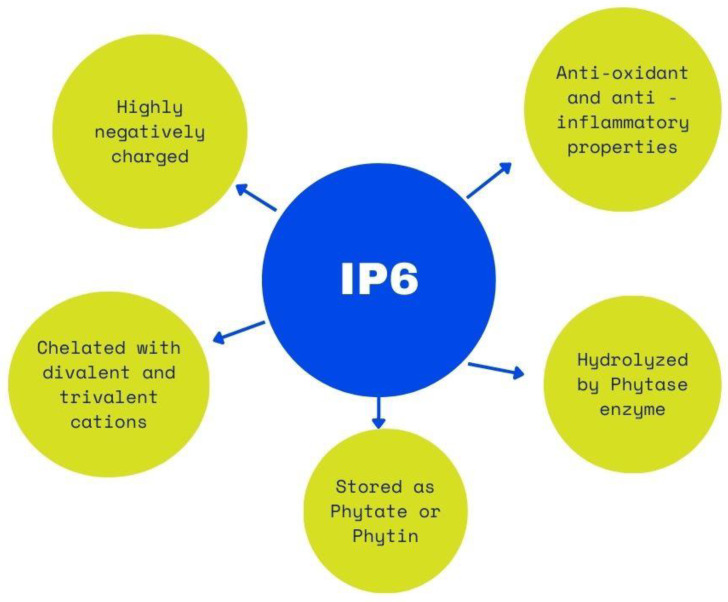
IP6 properties.

## Data Availability

No new data were created or analyzed in this study. Data sharing is not applicable to this article.

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
