# Peer review of "Application of Inositol Hexaphosphate and Inositol in Dental Medicine: An Overview"

_biomolecules, 2023, doi:10.3390/biom13060913_

Round 1
Reviewer 1 Report
This is a very well structured and clearly presented review. However, there is one aspect that would be important to clarify and comment on. Thus, throughout the review the bactericidal properties of phytate are indicated, which are probably detected because high concentrations of this product can be used at the oral level. Obviously for other biological fluids (blood, urine), since these high concentrations cannot be reached, these bactericidal effects would not be evidenced.
Author Response
Dear Mr/Mrs,
Thank you for your effort regarding our paper and your kind comments. Our review is based on the IP6 benefits in dental medicine. We presented the possible ways IP6 can be used in that specific field of medicine, and have revised our manuscript which is attached below.
We explained that our paper is based on IP6 usage in a narrow medicine branch – dental medicine, so we focused on IP6 properties when administered orally.
Regards,
Ana Druzijanic, PhD, DMD

Reviewer 2 Report
The manuscript titled, “Application of Inositol hexaphosphate and Inositol in Dental Medicine: An Overview” by Druzijanic et al.” summarizes beneficial effects of the biomolecule IP6 in various fields of dentistry. Overall, it is an interesting and well-written review. I have the following suggestions:
1 Line 69: “….IP6 also reacts with various enzymes….” IP6 has been shown to regulate AMPK mediated energy expenditure pathway (Zhu et al. JCI, 2016, 126, 4273-4288). This important function of IP6 should have been mentioned.
2. Line 79: In addition to energy metabolism, IP6 generated IP7 has many other functions, which have been reviewed (Chakraborty A. Biol Rev Camb Philos Soc., 2018, 93, 1203-1227, review). IP7’s functions should have been described in more detail.
3. Lines 21, 55, 89, 91, 276 describe beneficial (anti-inflammatory, anti-tumorigenic etc.) effects of IP6. Conversely, studies have shown IP6’s anti-nutrient effects. This view should also be discussed, and its potential impact on dental health should be mentioned.
4. In general, whether there are any potential concerns of using IP6 in dentistry should be discussed.
5. IP6 has been denoted as IP6 or IP-6 throughout the manuscript. The molecule should be represented consistently.
Author Response
Dear Mr/Mrs,
Thank you for your effort regarding our paper and your kind comments.
We have followed your recommendations, and enhanced our manuscript, which is attached below.
1st comment: We have explained the IP6 regulated AMPK-mediated energy expenditure pathway according to the advice.
2nd comment: We have added the recommended citation and expanded the section with recently published studies.
3rd comment: We followed the advice and incorporated a new section about the IP6 anti-nutrient effects.
4th comment: We have answered the comment in the manuscript conclusion.
5th comment: All three “IP-6” have been changed to “IP6” to be consistent throughout the manuscript.
Regards,
Ana Druzijanic, PhD, DMD

Reviewer 3 Report
Overall, although I found this manuscript to be an interesting topic, it is not suitable for publication in its present form. My main comments are listed below:
Firstly, the paper appears to have significant overlap with a previously published review article entitled “Phytic acid: Properties and potential applications in dentistry, 2021”. While the authors have included some new insights, I believe that the overlap with the previous review should be addressed in order to make the manuscript more original and distinct.
Furthermore, I have some concerns regarding the number of recent and updated articles cited in the paper. Upon reviewing the references section, I noticed that only 4 out of 27 cited articles are relatively recent and updated (say after 2021). I suggest that the author aim to provide a more diverse and up-to-date selection of articles to strengthen your arguments.
Lastly, while the authors do provide some valuable insights into the anti-inflammatory functions of IP6, I believe that the mechanism of how IP6 modulates immune cells could be described in more detail. It would be beneficial for you to expand on this aspect to provide a more comprehensive understanding of the topic.
Author Response
Dear Mr/Mrs,
Thank you for your effort regarding our paper and your kind comments.
We have followed your recommendations, and enhanced our manuscript, which is attached below.
1st comment: We agree that a significant part of the manuscript overlaps with the previous work, but we have also significantly expanded the contents and have included more recent studies and knowledge about the IP6.
2nd comment: We have updated the articles cited and have included the most recent systematic review about IP6 properties published in April 2023.
3rd comment: Following the instructions, we have added a section about IP6 modulation of immune cells, including the neutrophil and macrophage function modulation.
Regards,
Ana Druzijanic, PhD, DMD

Round 2
Reviewer 3 Report
After rechecking the document and verifying that the authors have implemented all the changes, the manuscript is considered ready for publication.